# The Production of Bioaroma by *Auriporia aurulenta* Using Agroindustrial Waste as a Substrate in Submerged Cultures

Rafael Donizete Dutra Sandes [1], Mônica Silva De Jesus [1], Hannah Caroline Santos Araujo [1], Raquel Anne Ribeiro Dos Santos [2], Juliete Pedreira Nogueira [1], Maria Terezinha Santos Leite Neta [1] and Narendra Narain [1,*]

1   Laboratory of Flavor and Chromatographic Analysis, Federal University of Sergipe, Av. Marechal Rondon, s/n, Jardim Rosa Elze, São Cristóvão 49100-000, Sergipe, Brazil; rafael.donizete.dutra@gmail.com (R.D.D.S.); monicasj.sst@gmail.com (M.S.D.J.); hcarol197@gmail.com (H.C.S.A.); juliete_nogueira@yahoo.com (J.P.N.); terezinhaleite@gmail.com (M.T.S.L.N.)

2   Federal Institute of Education, Science and Technology of Sergipe, Rod. BR 101, Km 96, s/n, Quissamã, São Cristóvão 49100-000, Sergipe, Brazil; eng.raquelanne@gmail.com

*   Correspondence: narendra.narain@gmail.com; Tel.: +55-793-194-6514

**Abstract:** The present study was carried out to investigate the potential of the basidiomycete *Auriporia aurulenta* to metabolize residues remaining from the processing of umbu, cajá, plum, and persimmon fruits for the production of natural aroma compounds using submerged fermentation. The volatile compounds obtained from the fermentation of *A. aurulenta* cultivated in these residues were extracted via stir bar sorptive extraction (SBSE) and analyzed using a gas chromatography-mass spectrometry (GC-MS) system. Esters and alcohols were the main compounds produced, with emphasis on the compounds 2-phenethyl acetate and 2-phenylethanol, which were mainly produced from umbu residue. The acid medium favored the production of 2-phenethyl acetate, reaching its maximum value (11.38 mg/L) on day 3.5, while higher concentrations of 2-phenylethanol were found in the basic medium, with optimal production (2.27 mg/L) on the 7th day. By varying the concentrations of pre-inoculum and residue in the optimization of this fermentation process, it was possible to double the production (24.47 mg/L) of 2-phenethyl acetate and obtain a seven times higher concentration (15.56 mg/L) of 2-phenylethanol. The diversity and expressive production of these aromatic compounds found in the fermentation media using these agroindustrial residues indicate that their use as substrates is an economical and environmentally viable alternative.

**Keywords:** agroindustrial residue; *Auriporia aurulenta*; bioaroma; 2-phenethyl acetate

## 1. Introduction

Currently, there is growing interest in and demand for the production of aroma compounds of natural origin for the food, cosmetics, and pharmaceutical industries. Biotechnological processes have become very attractive alternatives for obtaining these products naturally, called bioaromas, when they are generated from fermentation processes using microorganisms.

The microorganisms used in these bioprocesses are the basidiomycetes, which comprise almost all edible mushrooms and possess a particular extracellular enzymatic system that is able to synthesize various volatile compounds [1]. These fungi have been studied for the production of aromas in a liquid and submerged environment, for example, *Pleurotus sapidus* in the production of the terpene nootkatone; *Phanerochaete chrysosporium* and *Pycnoporus cinnabarinus* in the production of vanillin; *Ischnoderma benzoinum*, *Ischnoderma resinosum*, and *Bjerkandera adusta* in the production of benzaldehyde; and *Rhodotorula aurantiaca* and *Sporidiobolus ruinenii* in the production of γ-decalactone [2–12].

Basidiomycetes are also capable of producing bioaromas through the biotransformation of complex food by-products. The application of agroindustrial residues in bio-

processes is a way of using low-cost alternative substrates to supply natural aromas. In addition, it contributes to the preservation of the environment, making the production process of aromas economical and sustainable [13,14]. These residues are usually rich in nutrients such as lipids, amino acids, sugars, vitamins, mineral salts, and bioactive compounds, which can act as biogenetic precursors or a conducive means for microorganisms to produce aroma compounds through biotransformation or by synthesizing the new compounds [15–19].

Although there is a limited number of scientific articles in which fruit residues have been used as substrates for the production of odorous compounds with basidiomycetes, the results of these works have indicated that agroindustrial residues constitute a potential substrate for the generation of aroma compounds. These studies have focused on the use of coconut and pineapple residues, peanut shells, corn, and wheat bran for the production of vanillin via the basidiomycetes *Phanerochaete chrysosporium* and *Pycnoporus cinnabarinus* [10,20–24] and apple residue for the production of 3-phenylpropanal, 3-phenyl-1-propanol, and benzyl alcohol compounds via the basidiomycete *Tyromyces chioneus* [15]. However, there has been no scientific article published as of yet that has investigated the basidiomycete *Auriporia aurulenta* for the production of aroma compounds using agroindustrial residues. With the growing increase in the generation of agroindustrial residues from the production and processing of fruits and with the concern for the preservation of the environment, alternatives for the sustainable disposal of these materials have been widely investigated, and one of them is the use of these residues as a culture medium for microorganisms producing volatile aromatic compounds [25,26]. Thus, the present work investigates the potential of *Auriporia aurulenta* to metabolize residues of umbu (*Spondias tuberosa* L.), cajá (*Spondias mombin* L.), plum (*Prunus domestica* L.), and persimmon (*Diospyros kaki* L.) for the natural and sustainable production of aromatic compounds through submerged fermentation.

## 2. Materials and Methods

### 2.1. Chemical

Asparagine monohydrate and analytical grade standards (n-alkanes C8–C30, methyl nonanoate) were obtained from Sigma-Aldrich (Missouri, USA and Taufkirchen, Germany). Glucose was purchased from Neon Commercial Analytical Reagents Ltda. (Suzano, São Paulo, Brazil); potassium hydrogen phosphate was purchased from Qhemis, Scientific Hexis (Jundiaí, São Paulo, Brazil); magnesium sulfate, zinc sulfate, and eth-ylenediaminetetraacetic acid (EDTA) were purchased from Dinâmica Química Contemporânea Ltda. (Indaiatuba, São Paulo, Brazil); copper sulfate pentahydrate was acquired from Chemco Industry and Commerce Ltda. (Hortolândia, São Paulo, Brazil); 3,5-dinitrosalicylic acid (DNS) and ferric chloride were purchased from Vetec Fine Chemicals Ltda. (Duque de Caxias, Rio de Janeiro, Brazil); Mueller Hinton Agar was purchased from Kasvi Import and Distribution of Products for Laboratories Ltda. (São José dos Pinhais, Paraná, Brazil); and yeast extract (Merck KGaA, Darmstadt, Germany). All the other chemicals were of analytical grade or HPLC purity standard.

### 2.2. Agroindustrial Residue

The agroindustrial residues from the processing of umbu (*Spondias tuberosa* L.), cajá (*Spondias mombin* L.), plum (*Prunus domestica* L.), and persimmon (*Diospyros kaki* L.) fruits were donated by the company Pomar do Brasil Indústria e Comércio de Alimentos Ltda., Aracaju, Brazil. The residues were subjected to drying at 35 °C in an oven with air circulation (model MA 035/2, Marconi) until they reached a moisture content of less than 5%, when they were then crushed in an IKA mill and sieved through a 20-mesh sieve. The total reducing sugars were determined for the 4 residues using the DNS method [27].

### 2.3. Strains and Growing Conditions of Crops

The strain of the basidiomycete fungus *Auriporia aurulenta* was acquired from the library of the Institute of Food Chemistry at Leibniz University, Germany. All cultivation procedures were performed under sterile conditions. Stock cultures were maintained in Petri dishes containing SNL (Standard Nutrition Liquid) agar medium.

The pre-inoculum of *Auriporia aurulenta* was obtained following the methodology of Fraatz et al. [3]. In flasks, 150 mL of SNL culture medium, composed of 30 g/L of glucose; 4.6 g/L of asparagine monohydrate; 3.0 g/L of yeast extract; 1.6 g/L of potassium hydrogen phosphate; 0.6 g/L of magnesium sulfate; and 0.5 mL/L of trace element solution (solution to supplement the medium prepared with the following reagents: 5 mg/L of copper sulfate pentahydrate; 80 mg/L of ferric chloride; 30 mg/L of magnesium sulfate; 90 mg/L of zinc sulfate; and 400 mg/L of EDTA), was added. In the sterile SNL broth, with a pH of 6 determined by a pHmeter (Hanna, model HI2210), a $1 \times 1$ cm portion of the microorganism was dispersed using an Ultra Turrax (Heidolph, model SilentCrusher M). The pre-inoculum was kept in a shaker incubator until development for 15 days at 24 °C and 150 rpm, protected from light.

### 2.4. Submerged Fermentation

The fermentation tests were carried out using agroindustrial residues as substrates, following the methodology of Bosse [15] and Grosse [28] and using the SNL minimum culture medium (consisting of reduced component concentrations of 10 g/L of glucose; 3 g/L of asparagine monohydrate; 1 g/L of yeast extract; 1.5 g/L of potassium hydrogen phosphate; 0.5 g/L of magnesium sulfate; and 1 mL/L of trace element solution). For the fermentation process, 25 mL of *A. aurulenta* pre-inoculum was added to 250 mL erlenmeyer flasks containing 125 mL of SNL minimum with the addition of 5% (6.25 g) of the agroindustrial residues. Later, the pH was adjusted to 6 and the flasks were kept under agitation (150 rpm) in a shaker for 7 days at 24 °C and protected from light. Parallel to the fermentation with the residue, a control culture in SNL minimum was also prepared.

#### Kinetics and Optimization of the Fermentation Process for the Production of Aroma Compounds

The kinetic study of the fermentation process for the production of the main aromatic compounds, using the residue with the highest production of these compounds as substrate, was carried out under the same conditions as in the previous item, being monitored for 21 days with the analysis being performed every 3.5 days (0; 3.5; 7; 10.5; 14; 17.5; and 21 days), using initial pHs of 6 and 3. The reduction in total sugars was also monitored in these time intervals, using the method described by Miller [27].

The optimization of the fermentation conditions was investigated on the days of highest production, which were determined in the kinetic study. The effects of the pre-inoculum (X) and residue (Y) quantitative variables were evaluated using a Rotational Central Compound Design (DCCR), consisting of a 22 factorial design, 4 axial points, and 3 central point replications, totaling 11 attempts. Table 1 shows the range of variables studied and the corresponding coded levels. In order to maximize the use of residue and minimize the amount of pre-inoculum from the previous fermentation, the values of 6.25 g and 25 mL were considered for variables X (−1.41) and Y (+1.41), respectively.

**Table 1.** Levels of independent variables (X and Y) for Rotational Central Composite Design.

| Independent Variables | −1.41 | −1 | 0 | 1 | 1.41 |
|:---:|:---:|:---:|:---:|:---:|:---:|
| X (mL) | 8.00 | 13.12 | 16.50 | 21.56 | 25.00 |
| Y (g) | 6.25 | 9.04 | 13.12 | 17.22 | 20.00 |

### 2.5. Analysis of Volatile Compounds

The volatile compounds present in the submerged cultures were extracted using stir bar sorptive extraction (SBSE). For the extraction, 10 mL of filtered fermentative broth

and 1 g of sodium chloride (NaCl) were placed in a 20 mL vial sealed with a septum, equilibrated at a temperature of 40 °C in a water bath under magnetic stirring at 1000 rpm. The PDMS twister was immersed in the sample for the adsorption of the analytes for 60 min and, after extraction, it was removed with tweezers, rinsed with deionized water, dried with lint-free tissues, and inserted into a glass tube conditioned for the Thermal Desorption Unit bar (TDU 2; Gerstel, Germany). All extractions were performed in triplicate.

The aroma compounds were desorbed in the system using a TDU (Thermal Desorption Unit; Gerstel, Germany) with the following temperature settings: 30 °C for 0.5 min, which then increased to 250 °C at 120 °C/min and was maintained at this temperature for 5 min to desorb the aromatic compounds (thermodesorption). The analytes were transferred to a cold injection system (CIS) equipped with a glass wool liner (2 mm; Gerstel). After thermodesorption, the CIS was immediately heated at 12 °C/s from 45 °C to 250 °C, where it was held for 5 min to release the analytes into the gas chromatograph column.

### 2.5.1. Separation of Volatile Compounds

The volatile compounds were analyzed on a gas chromatograph (model 7890B; Agilent Technologies, Santa Clara, CA, USA) coupled to a mass spectrometer (model 5977A, Agilent Technologies). Separation was achieved on an HP-5MS capillary column (30 m × 0.25 mm id, 0.25 μm film thickness) purchased from J&W Scientific, Agilent Technologies (Santa Clara, CA, USA). The injector temperature was set at 240 °C and helium was used as a carrier gas at a flow rate of 1.3 mL per min in the splitless injection system. The conditions used in the GC system were: an initial oven temperature of 40 °C, which increased at a rate of 3 °C/min to 130 °C, where it remained for 1 min, and later at the rate of 15 °C/min to 250 °C (1 min). The transfer line, ion source, and quadrupole temperatures were 260 °C, 280 °C, and 180 °C, respectively. The ionization source voltage was 70 eV, with the mass scanning range varying between 35 and 350 amu.

### 2.5.2. Identification of Volatile Compounds

The volatile compounds were identified by comparing their mass spectra with those obtained from the NIST library (National Institute of Standards & Technology, Gaithersburg, MD, USA) and comparing the linear retention indexes (LRI) of the compounds, which were calculated based on the retention times of a series of n- alkanes (C8–C30) under identical analytical conditions, with those from the literature articles and NIST databases.

An internal standardization method was used to quantify the identified volatile compounds. The concentrations of the identified compounds were calculated from the ratio between the peak area and the peak area of the internal standard, expressed in mg/L. In this study, methyl nonanoate was used as an internal standard, with a final concentration of 15 μg/mL.

### 2.6. Statistical Analysis

The data obtained were subjected to an Analysis of Variance (ANOVA) and a Tukey's Test was performed to determine the significant differences between the samples at a 5% significance level ($p \leq 0.05$), using the statistical program XLSTAT (Addinsoft Inc., Paris, France, FR, 2016). For the optimization of the fermentation process, the DCCR and Response Surface Methodology were analyzed using the STATISTICA program version 12.0.

## 3. Results
### 3.1. Volatile Profile of Fermented Products

The residues of umbu, cajá, persimmon, and plum fruits were used as substrates for the cultivation of the basidiomycete *Auriporia aurulenta* and investigated for the production of aroma compounds via submerged fermentation. The data on the compound names, their retention indices—calculated and from the literature—concentrations, and odor characteristics of the volatile compounds identified and quantified in each fermentation process are presented in Table 2. A total of 46 compounds were identified and quantification in the

submerged culture broth, with a predominance of esters (25 compounds) and alcohols (13), as well as terpenoids (4), aldehydes (2), ketones (1), and lactones (1).

**Table 2.** Volatile compounds produced (mg/L) by *Auriporia aurulenta* using umbu, cajá, persimmon, and plum residues as substrates, in a submerged fermentation process for 7 days.

| Compounds | RI+ | RI* | Umbu Residue | Cajá Residue | Persimmon Residue | Plum Residue | Control | Odor Description |
|---|---|---|---|---|---|---|---|---|
| 2-Methyl-1-propanol | 618 | 618 | 0.87 a | 0.68 b | 0.50 c | 0.41 d | 0.91 a | Wine |
| 2-Methyl-1-butanol | 737 | 736 | 0.69 b | 0.28 c | 0.20 d | 0.07 e | 1.07 a | Alcoholic, fruity, wine |
| 3-Methyl-1-butanol | 765 | 768 | 2.66 a | 0.88 d | 1.83 b | 0.80 d | 1.63 c | Fermented, banana |
| 2-Methylpropyl acetate | 772 | 772 | 0.06 a | | | | 0.03 b | Sweet, fruity, banana |
| Isoamyl acetate | 876 | 877 | 0.80 a | 0.57 b | 0.23 c | 0.20 c | 0.82 a | Sweet, fruity, banana |
| 2-Methylbutanol acetate | 879 | 879 | 0,13 a | | | | 0.03 b | Fruity, sweet, banana |
| Prenyl acetate | 910 | 909 | 0,01 c | | 0.02 b | 0.04 a | | Banana, fruity, jasmine |
| Benzaldehyde | 962 | 962 | 0.03 b | 1.27 a | | | | Fruity, almond |
| 1-Octen-3-ol | 980 | 980 | 0.07 b | 0.01 d | 0.01 d | 0.12 a | 0.03 c | Green, fungus, mushroom |
| 3-Octanone | 987 | 984 | 0.19 a | 0.19 a | 0.14 b | 0.14 b | 0.09 c | Moldy, mushroom |
| Butyl butanoate | 995 | 995 | | | 0.03 | | | |
| 3-Octanol | 996 | 996 | 0.13 b.c | 0.15 a | 0.12 c | 0.14 b | 0.06 d | Woody, mushroom |
| (Z)-3-Hexenyl acetate | 1008 | 1005 | 0.46 a | | | 0.09 b | | Sweet, fruity, banana |
| Hexyl acetate | 1015 | 1015 | | 0.22 a | 0.03 b | 0.03 b | | Fruity, banana |
| (E)-2-Hexenyl acetate | 1018 | 1007 | 0.10 | | | | | Apple, banana |
| Benzyl alcohol | 1037 | 1038 | 0.61 d | 2.07 a | 0.64 c,d | 1.68 b | 0.76 c | Fruity, floral, candy |
| (Z)-3-Octenol | 1050 | 1047 | | 0.02 | | | | Fresh, herbal, melon |
| (E)-2-Octenol | 1065 | 1069 | | 0.02 c | 0.16 a | 0.15 b | 0.01 d | Vegetable, citrus, green |
| 1-Octanol | 1071 | 1070 | 0.06 c | 0.07 c | 0.25 a | 0.19 b | 0.07 c | Citric, mushroom |
| (Z)-5-Octenol | 1071 | 1051 | 0.10 | | | | | Green, mushroom |
| Linalool | 1100 | 1099 | 0.14 b | 0.01 c | | | 0.31 a | Pink, floral, orange |
| Heptyl acetate | 1112 | 1114 | 0.04 | | | | | Woody, apricot, pear |
| 1-Octenyl-3-acetate | 1113 | 1113 | | | 0.21 | | | Fruity, green, herbal |
| 2-Phenylethanol | 1118 | 1112 | 2.25 a | 0.24 c | 0.01 d | 0.77 b | 0.27 c | Floral, roses |
| Benzyl acetate | 1169 | 1162 | 0.68 c | 1.32 b | 0.15 d | 0.16 d | 1.41 a | Floral, fruity, jasmine |
| Methyl salicylate | 1199 | 1200 | | | | 0.06 | | Slightly, phenolic, mint |
| Octyl acetate | 1235 | 1220 | 0.01 d | 0.24 a | 0.03 c | 0.09 b | | Floral |
| 3-Phenylpropanol | 1254 | 1253 | 0.37 b | 0.60 a | | 0.04 d | 0.13 c | Spicy, sweet |
| Heptyl butanoate | 1273 | 1275 | | 0.07 | | | | Fruity, floral, green, tea |
| 2-Phenethyl acetate | 1277 | 1271 | 5.07 a | 0.84 b,c | 0.77 b | 1.24 b | 0.54 c | Fruity, honey, rose, floral |
| γ-Octalactone | 1278 | 1277 | 0.37 a | 0.09 b | 0.01 c | 0.01 c | 0.10 b | Sweet, coconut |
| (E)-Cinnamaldehyde | 1280 | 1283 | | 0.06 | | | | Sweet, cinnamon |
| p-Cymen-7-ol | 1295 | 1295 | 0.18 | | | | | Cumin, spicy, herbs |
| Myrtenyl acetate | 1299 | 1305 | | | 0.05 a | | 0.01 b | Fruity, sweet, herbal |
| Cinnamyl alcohol | 1307 | 1312 | | 0.39 | | | | Sweet, green, hyacinth |
| 1,4-p-Menthadien-7-ol | 1320 | 1315 | 0.16 | | | | | |
| Benzyl butanoate | 1343 | 1345 | | 0.06 | | | | Apricot, fruity, jasmine |
| 2-Phenylethyl propanoate | 1350 | 1350 | 0.02 | | | | | Honey, fruity, floral |
| 3-Phenylpropyl acetate | 1371 | 1373 | 0.68 b | 0.21 c | 0.04 e | 0.11 d | 0.77 a | Spicy, cinnamon |
| Methyl cinnamate | 1383 | 1380 | | 0.02 a | | 0.03 a | | Sweet, balsamic |
| (E)-2-Hexenyl hexanoate | 1384 | 1375 | | 0.03 | | | | Herbal, green |
| (E)-β-Damascenone | 1386 | 1382 | 0.18 | | | | | Fruity, honey, rose, apple |
| Cuminyl acetate | 1422 | 1432 | 0.40 | | | | | Fresh, fruity, sweet, herbs |
| Cinnamyl acetate | 1451 | 1453 | 0.33 b | 0.41 a | | | | Cinnamon, floral, sweet |
| Ethyl (E)-cinnamate | 1456 | 1463 | | 0.04 | | | | Floral, honey |

Values in the same row with different superscripts are significantly different ($p \leq 0.05$); RI+—Mean of the calculated Linear Retention Index; RI*—Linear retention index according to the literature (NIST, 2014).

The *Auriporia aurulenta* cultivated at the SNL minimum (control) produced 20 volatile compounds. The major compounds were 3-methyl-1-butanol (1.63 mg/L), benzyl acetate (1.41 mg/L), 2-methyl-1-butanol (1.07 mg/L), 2-methyl-1-propanol (0.91 mg/L), isoamyl acetate (0.82 mg/L), and 3-phenylpropyl acetate (0.77 mg/L). Most of these compounds possess floral, fruity, and sweet aroma characteristics. All these compounds showed a decrease in their concentrations in the fermented products, except for 3-methyl-1-butanol, whose concentration was higher in the products fermented with the umbu and persimmon residues, presenting a significant difference ($p \leq 0.05$) from the other products.

Table 2 shows that larger numbers of volatile compounds were obtained in the fermentation with umbu and cajá residues being used as substrates, with 31 and 29 being obtained, respectively, while the fermentations with plum and persimmon residues presented only

22 compounds. Among the total number of compounds, twenty-seven were found only in the fermented products, which were absent in the control unfermented material. Among these compounds, there was the production of benzaldehyde, cinnamyl acetate, cinnamyl alcohol, octyl acetate, and hexyl acetate using cajá residue; 3-(Z)-hexenyl acetate, cuminyl acetate, (E)-β-damascenone, p-cymen-7-ol, 1,4-p-menthadien-7-ol, and (E)-2-hexenyl acetate using umbu residue; and 1-octenyl-3-acetate using persimmon residue.

*Auriporia aurulenta* using agroindustrial residues produced several volatile compounds in expressive concentrations. Esters were the main compounds produced in the fermentations, highlighting those with a production greater than 0.1 mg/L: 2-phenethyl acetate, 3-(Z)-hexenyl acetate, cuminyl acetate, cinnamyl acetate, 2-methylbutyl acetate, (E)-2-hexenyl acetate, octyl acetate, 1-octenyl-3-acetate, and hexyl acetate. Among these compounds, 2-phenethyl acetate was mainly produced by all the residues, with the highest production (5.07 mg/L) occurring with the umbu residue, significantly different ($p \leq 0.05$) in relation to the others. In fact, it was observed that the product fermented with this residue showed higher values for most of these compounds compared to the data obtained by using the other residues. Medeiros et al. [29] also found esters and alcohols to be the major classes when using cassava bagasse fermented by *Kluyveromyces marxianus*.

Alcohols were the second major class of volatile compounds. The 8-carbon alcohols are well known as fungal volatile organic markers. Among them, the alcohols 1-octen-3-ol, 3-octanol, 1-octanol, and (Z)-5-octenol are considered as the main compounds responsible for mushroom odor. [30]. In addition to these, the production of 2-phenylethanol (2.25 mg/L) was highlighted, mainly occurring with umbu residue and significantly different ($p \leq 0.05$) in relation to the other residues, considered as one of the most important aromatic alcohols. Additionally noteworthy was benzyl alcohol (2.07 mg/L), produced mainly in the fermented products with cajá residue, as well as benzaldehyde (1.27 mg/L), which showed a significant difference ($p \leq 0.05$) in relation to the other fermented products. Previous studies have reported a direct relationship between benzyl alcohol and benzaldehyde, in which basidiomycetes use benzyl alcohol as a precursor to benzaldehyde [11]. Other studies have shown that benzyl alcohol can be formed by the reduction in benzaldehyde during the fermentation process with basidiomycetes [31], and interconversions between these compounds were observed by Kawabe [32], where benzaldehyde produced by fermentation pathways was reduced to benzyl alcohol and slowly converted back into benzaldehyde.

The addition of umbu residue as a substrate for *A. aurulenta* led to the production of some volatile compounds that were not found in the fermentations with other residues. It was observed that, of the total of four terpenoid compounds present in the fermented products, three of these were produced exclusively using umbu residue: (E)-β-damascenone (0.18 mg/L); p-cymen-7-ol (0.18 mg/L); and 1,4-p-menthadien-7-ol (0.16 mg/L). In addition, linalool (0.14 mg/L) was preferably produced using umbu residue but was also present in the fermentation used with cajá residue (0.01 mg/L). Among these, the importance of linalool stands out. It is a prevalent component in vegetable essential oils and commonly used in the fragrance and pharmaceutical industry, also presenting anti-inflammatory and analgesic effects. In addition, (E)-β-damascenone, an important odorous compound with "fruity" notes that contributes to the aroma of many food products, was noted.

The only lactone quantified in this study, γ-octalactone, showed a significant increase ($p \leq 0.05$) during the fermentation with umbu residue, with a production of 0.37 mg/L, while the fermentations with other residues showed decreases in their concentrations in relation to the control. γ-Octalactone is a fragrance compound with a sweet coconut aroma, often used in perfumery for floral fragrances and also as a food flavoring, being an aromatic component of beer and brandy. Although it was not a major compound in the present study, the production of γ-octalactone in the product fermented with *A. aurulenta* and umbu residue was higher than that reported by Wickramasinghe [11] in the fermentation broth of the edible mushroom *Ischnoderma resinosuma* (14.9 μg/kg).

### 3.2. Kinetics of Formation of Aroma Compounds

The change in volatile composition that occurred during the fermentation process was analyzed and the selection of the compounds of interest was carried out considering the concentration and aromatic potential of the compounds produced. Thus, the compounds 2-phenethyl acetate and 2-phenylethanol were selected because they were produced in higher concentrations by *A. aurulenta* using umbu residue, as was (*E*)-*β*-damascenone because it is a potent odorous compound with a low odor threshold and is very important for the flavor industry, although it was produced in a low concentration. Thus, the kinetics of the formation of these compounds were monitored for 21 days in submerged cultures of *A. aurulenta* using umbu residue as substrate, in acid medium (pH 3) and in basic medium (pH 6). An erlenmeyer flask was collected every three and a half days to determine the volatiles, pH, and total sugar concentrations.

The umbu residue showed a small amount of total sugars (0.17 g/L), so the sugar available in the fermentation process refers to that present in the SNL minimum, which was reduced in a non-linear way during the fermentation process. After inoculation, a 2-day lag phase was observed in the basic medium and subsequently, so was a drastic decrease in the sugar from 11.20 g/L to 2.69 g/L in just 3 days and, parallel to this period, there was an increase in the production of 2-phenylethanol (Figure 1). In the acidic medium, however, a higher consumption of sugar was observed between the seventh and tenth and a half day, the period in which the highest amount of 2-phenethyl acetate was produced.

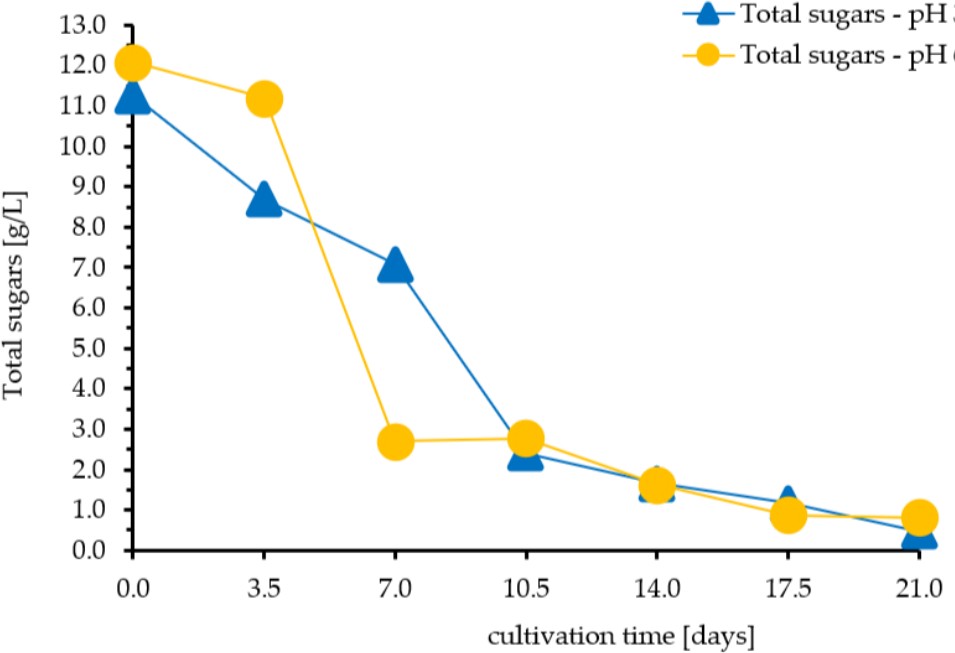

**Figure 1.** Total sugar concentration during 21 days of submerged cultivation of *A. auruenta* with umbu residue at pH 3 and pH 6.

The low sugar content of the umbu residue may have favored the high yield of 2-phenethyl acetate, and its highest production (11.39 mg/L) was reached on day 3.5 at a pH of 3, with a significant decrease occurring after day 7, while at pH 6, this production was only 3.88 mg/L (Figure 2A). The behavior of *A. aurulenta* at a pH close to 6 favored the biosynthesis of 2-phenylethanol alcohol, reaching its highest production (2.28 mg/L) on the seventh day, while at pH 3, the production was only 1.31 mg/L (Figure 2B). As observed, it can be inferred that the microorganism *A. aurulenta*, using umbu residue as substrate, preferentially biosynthesizes the 2-phenethyl acetate ester in an acid medium and the 2-phenylethanol in a basic medium.

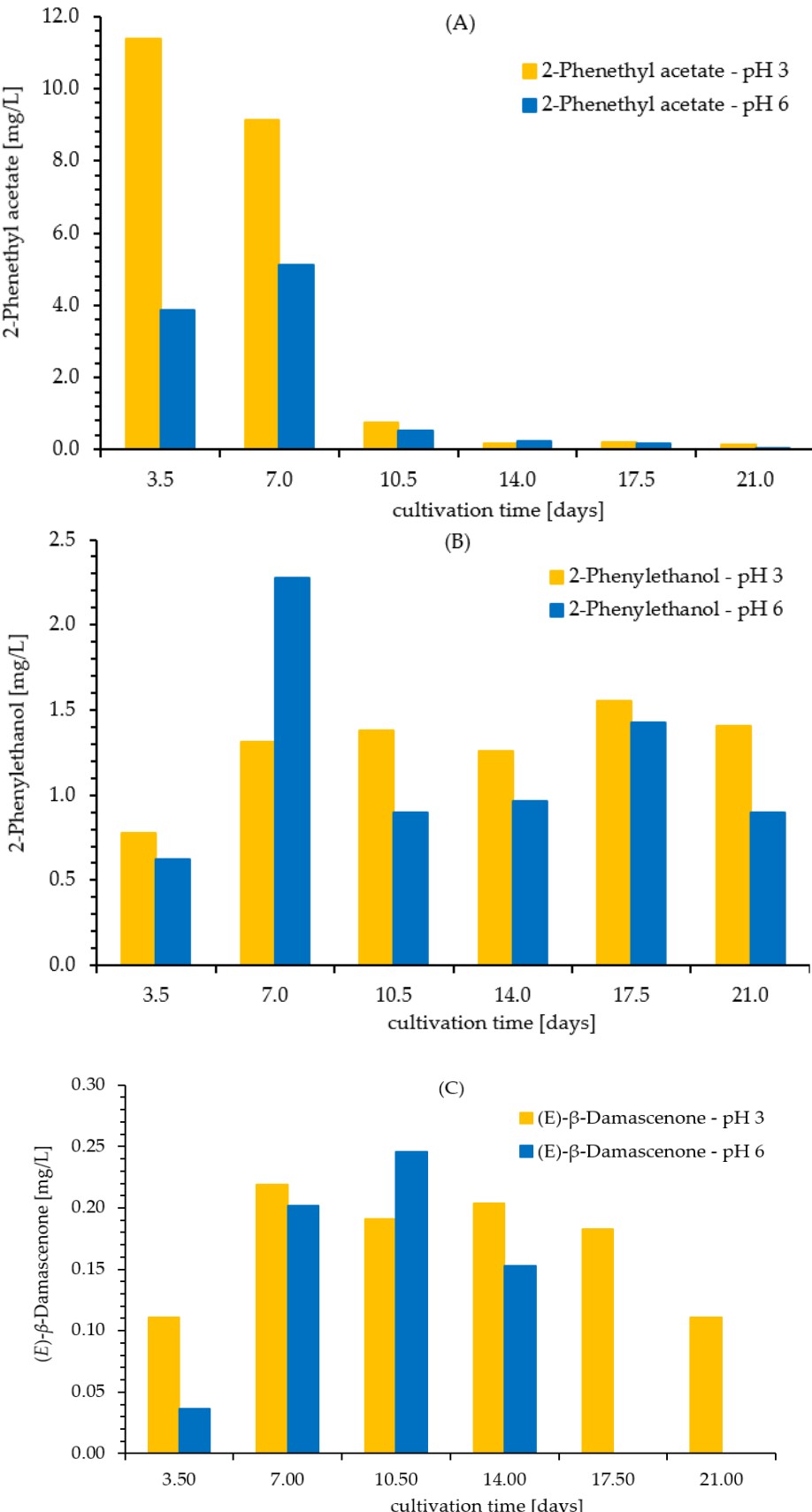

**Figure 2.** Kinetics of 2-phenethyl acetate (**A**), 2-phenylethanol (**B**), and (*E*)-*β*-damascenone (**C**) production during submerged cultivation of *A. aurulenta* with umbu residue at pH 3 and pH 6.

The production of (*E*)-$\beta$-damascenone reached a maximum content of 0.25 mg/L on day 10.5 at pH 6 (Figure 2C). However, considerable concentrations were also observed on the seventh day of fermentation, at both pHs. The increase in (*E*)-$\beta$-damascenone production from day 7 to day 10.5 was 18%, but this difference does not justify extending the time of the fermentation process; however, its production was more advantageous on the seventh day.

### 3.3. Fermentation—Process Optimization

A DCCR $2^2$ statistical experimental design was carried out to evaluate the effect of pre-inoculum and umbu residue on the production of 2-phenethyl acetate, 2-phenylethanol, and (*E*)-$\beta$-damascenone. Based on the results obtained in the study of the kinetics of the formation of these compounds, the time (d) and initial pH of the submerged fermentation were set at 3.5 and 3.0 for the production of 2-phenethyl acetate; 7 and 6.0 for the 2-phenylethanol production; and 7 and 3.0 for the (*E*)-$\beta$-damascenone production, respectively. The values of the independent variables (pre-inoculum and residue concentrations) and the experimental and predicted responses for each assay are shown in Table 3.

**Table 3.** Effect of pre-inoculum and residue concentrations on the production of volatile compounds obtained in submerged fermentation with *A. aurulenta*.

| Assay | Pre-Inoculum (mL) | Residue (g) | Experimental Values | | | Predicted Values | |
|---|---|---|---|---|---|---|---|
| | | | 2-Phenethyl Acetate (mg/L) | 2-Phenylethanol (mg/L) | (E)-$\beta$-Damascenone (mg/L) | 2-Phenethyl Acetate (mg/L) | 2-Phenylethanol (mg/L) |
| 1 | 11.44 | 9.04 | 7.32 | 5.41 | 0.19 | 4.00 | 4.80 |
| 2 | 21.56 | 9.04 | 3.57 | 6.71 | 0.18 | 4.00 | 8.93 |
| 3 | 11.44 | 17.22 | 13.05 | 2.86 | 0.23 | 13.43 | 4.80 |
| 4 | 21.56 | 17.22 | 9.28 | 6.25 | 0.47 | 13.43 | 8.93 |
| 5 | 8.00 | 13.12 | 8.25 | 8.22 | 0.24 | 5.78 | 3.97 |
| 6 | 25.00 | 13.12 | 10.10 | 15.56 | 0.23 | 5.78 | 10.91 |
| 7 | 16.5 | 6.24 | 4.23 | 6.65 | 0.23 | 6.14 | 6.55 |
| 8 | 16.5 | 20.00 | 24.47 | 9.90 | 0.19 | 21.96 | 6.55 |
| 9 | 16.5 | 13.12 | 4.10 | 4.15 | 0.22 | 5.78 | 6.55 |
| 10 | 16.5 | 13.12 | 4.11 | 4.67 | 0.24 | 5.78 | 6.55 |
| 11 | 16.5 | 13.12 | 3.41 | 4.68 | 0.26 | 5.78 | 6.55 |

The production values of 2-phenethyl acetate ranged from 3.41 to 24.47 mg/L and its highest production was found in test 8, where fermentation occurred with 16.5 mL of pre-inoculum and the maximum amount of residue (20 g). Keeping the same pre-inoculum concentration and decreasing the amount of residue, a drastic reduction in the production of this compound was observed, reaching a minimum value of 3.41 mg/L. Thus, it was verified that the increase in the concentration of residue caused the increase in the production of 2-phenethyl acetate; however the same relation was not observed for the concentration of pre-inoculum. When the highest concentration of pre-inoculum (25 mL) was used, there was a reduction in the production of 2-phenethyl acetate (10.10 mg/L).

Contrary to the behavior of the 2-phenethyl acetate production, 2-phenylethanol production was favored by increasing the pre-inoculum concentration (12.5 mL), reaching a value of 15.56 mg/L. However, when using the maximum amounts of residue (20 g) and lower values of pre-inoculum, there was a decrease in this production (9.90 mg/L). Regarding (*E*)-$\beta$-damascenone, its production ranged from 0.19 to 0.47 mg/L, with the maximum value being observed when 10.78 mL of pre-inoculum and 8.61 g of residue were used.

To determine the significant factors, the Pareto chart (Figure 3) was used, in which it was possible to verify that the linear (L) and quadratic (Q) effects of the residual variable were significant ($p \leq 0.05$) for the production of 2-phenethyl acetate. For the 2-phenylethanol production, only the quadratic effect (Q) of the pre-inoculum variable was significant ($p \leq 0.05$), while none of the variables were significant ($p > 0.05$) for the production of (*E*)-$\beta$-damascenone within the range studied. After eliminating the factors whose effects were not significant, statistical models were generated for the compounds 2-phenethyl acetate and 2-phenylethanol.

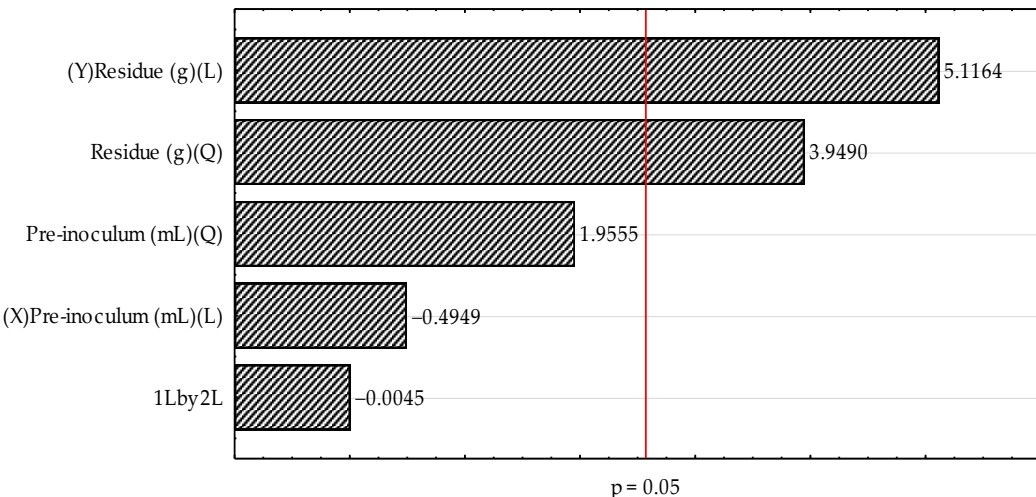

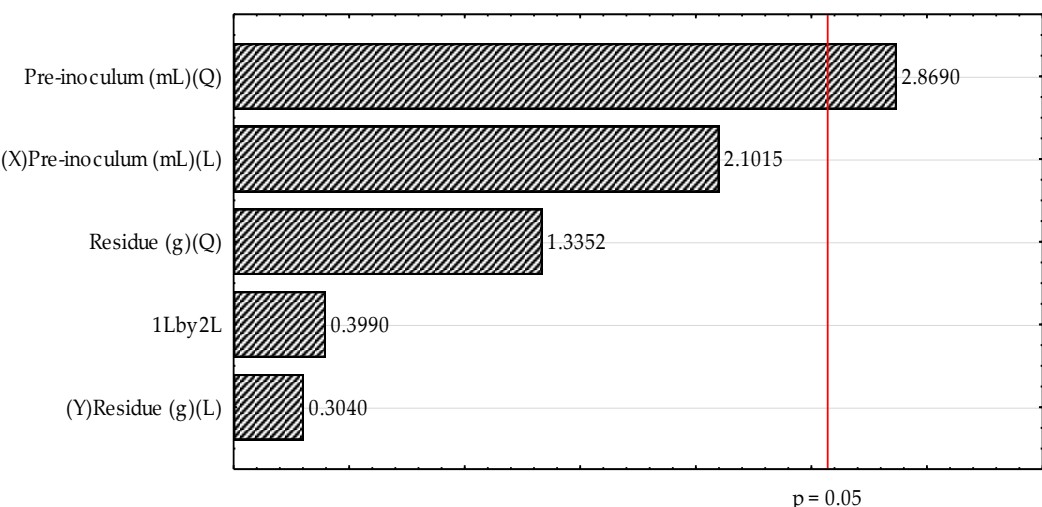

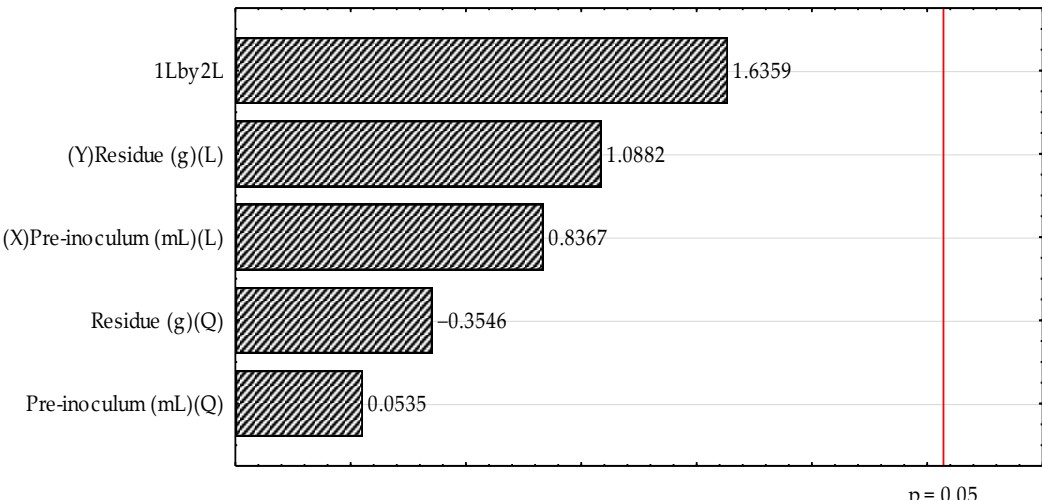

**Figure 3.** Pareto diagram for the effects of pre-inoculum and residue on the concentration of 2-phenethyl acetate, 2-phenylethanol, and (*E*)-*β*-damascenone.

The influence of the independent variables studied in the fermentation process for the production of 2-phenethyl acetate and 2-phenylethanol compounds can be better observed in the individual graphical representation of the statistical models, through the response surfaces (Figure 4), which were statistically significant ($p \leq 0.05$). For 2-phenethyl acetate, this statistical model ($Z = 0.759 - 3.432Y + 0.175Y^2$; where Z = the concentration of the compound produced and Y = the amount of residue) was considered valid for prediction purposes, as it presented a coefficient of determination ($R^2$) that was satisfactory or equal to 0.81, meaning that 81% of the data variation can be explained by this model. The statistical model determined for 2-phenylethanol ($Z = 3.180 + 0.012X^2$, where X = amount of pre-inoculum), although significant, presented a coefficient of determination ($R^2$) of only 0.36, explaining only 36% of the data variation. In these graphs, it is easy to see the conditions that maximized the production of each volatile compound, and that the maximums of all these models were obtained using different values of the studied factors.

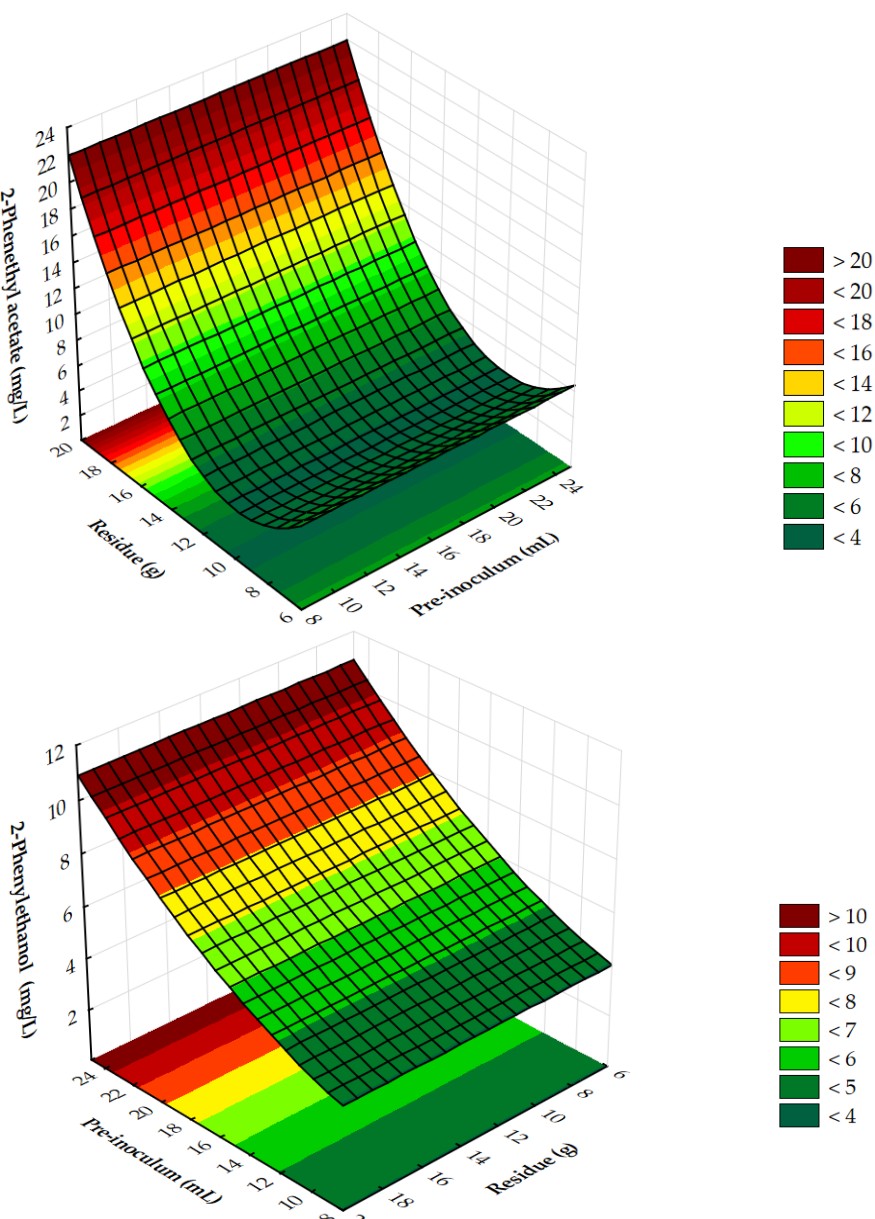

**Figure 4.** Response surface graphs for the effects of pre-inoculum and residue on the concentration of 2-phenethyl acetate and 2-phenylethanol.

The response surface presented in Figure 4, generated by the linear and quadratic model, indicated that the amount of residue positively influenced the production of 2-phenethyl acetate. This influence can be seen when observing the optimal region represented in red, that is, when using greater amounts of umbu residue, in the range between 19 g and 22 g, regardless of the amount of pre-inoculum used, it was possible to obtain higher concentrations of this compound. It was also observed that the pre-inoculum variation in the range studied did not impact this production. Regarding 2-phenylethanol, the response surface and contour indicated that its greater production occurred with an increase in the pre-inoculum concentration, reaching maximum values in the range above 22 mL of pre-inoculum.

The study of the amount of residue as a substrate and pre-inoculum is of great importance, since these parameters can influence the quantity and quality of the compounds generated, as well as the production time, impacting the higher or lower cost of the process. The results indicated great and promising perspectives for the scale-up of the aroma production process via fermentation in a submerged state using *A. aurulenta* with agroindustrial residues as substrates.

## 4. Discussion

The volatile profiles of the fermented broths produced by *Auriporia aurulenta* using residues of umbu, cajá, persimmon, and plum as substrates revealed a great complexity in the mixture of aroma compounds. The umbu residue stood out among the others in its expressive production of 2-phenethyl acetate and 2-phenylethanol, in addition to compounds such as 3-(Z)-hexenyl acetate, cuminyl acetate, $\gamma$-octalactone, (E)-$\beta$-damascenone, and *p*-cymen-7-ol.

(E)-$\beta$-damascenone, which can be considered a product of carotenoid oxidative degradation [33], was one of the terpenoids present in umbu residue and contributes to the aromas of various products such as green tea [34], wines [31,35,36], Arabica coffee [37], buckwheat honey [38], and in liquid malt extract for the bakery industry [39]. The production of (E)-$\beta$-damascenone by basidiomycetes has been investigated and, so far, the concentrations obtained in these studies have been relatively small, ranging from 0.004 to 0.16 mg/L [34,40], in comparison to the present study, where maximum production was 0.47 mg/L.

Work with basidiomycetes is more focused on the production of terpenes [28,41,42] and vanillin [9,22,43–45], however, *Auriporia aurulenta* has shown potential for the synthesis of esters with different aromas (peach, pineapple, banana, citrus, and rose), depending on the growing conditions. Mantzouridou [46] obtained a higher production of volatile esters by yeast using orange peel as a substrate. *Ceratocystis fimbriata* (ascomycete fungi) also increased the production of esters with a fruity aroma using 50% soybean meal residue as a nitrogen source and 25% sugarcane molasses as a carbon source [47].

*A. aurulenta* produced significant amounts of 2-phenethyl acetate, a compound with floral, rose, and honey aromatic notes, showing statistically significant ($p \leq 0.05$) differences between umbu residue and the other substrates. This compound was also produced by the basidiomycetes *Antrodia camphorata* and *Polyporus tuberaster*, but represented lower values [48,49]. Gallois et al. [50] studied 29 basidiomycetes for the production of aroma compounds, and only the strains *Lenzites trabea* and *Trametes pini* managed to produce 2-phenethyl acetate, with concentrations ranging from 0 to 0.05 mg/L. These concentrations were lower than those obtained in our study, where the variations ranged from 0.84 mg/L (persimmon residue) to 5.07 mg/L (umbu residue) for 7 days of fermentation.

2-Phenylethanol can be formed via synthesis by basidiomycetes or through the bioconversion of appropriate precursors such as *L*-phenylalanine [51]. In our study, using umbu residue, the preliminary concentration of 2-phenylethanol was 2.25 mg/L after 7 days of fermentation. Previous papers have also reported the production of 2-phenylethanol by basidiomycetes such as *Vararia effuscata*, *Fomes annosus*, *Lenzites betulina*, *Phellinus robustus*, *Polyporus frondosus*, *Poria subacida*, *Poria subvennispora*, and *Annillaria ostoyae*, reaching

mean concentration values of 0.25 to 1.00 mg/L [50,52,53]. *Torulaspora delbrueckii* had the highest sugar consumption and highest production of higher alcohols (isoamyl alcohol, 2-phenylethanol) and ethyl esters (ethyl octanoate and ethyl decanoate). *Williopsis saturnus* could indeed consume more nitrogen sources and produce greater amounts of acetate esters, but the ethyl acetate produced was at an excessive level [54].

Basidiomycetes were able to produce aroma compounds in a short period of time, ranging from 3.5 to 7 days. The 2-phenethyl acetate showed the highest production at 3.5 days, and the optimization of the fermentation process made it possible to obtain concentrations of up to 24.47 mg/L of this ester. The large increase in the concentration of 2-phenethyl acetate was directly related to the greater amount of umbu residue, suggesting that it is an adequate source of nitrogen for its production. The production of 2-phenylethanol on the 7th day by *A. aurulenta* is a common feature of most basidiomycetes, reaching a maximum value of 15.56 mg/L in the optimization process, which was directly related to the amount of the microorganism in the medium.

The formation of aroma compounds in this study may be related to the available nutrients and precursors present in the residues, which constitute a suitable environment for the fermentation process for basidiomycetes. Based on the results obtained, it could be concluded that *Auriporia aurulenta* is a promising basidiomycete in the production of esters using agroindustrial residues as substrates.

## 5. Conclusions

The aroma compounds present in the fermented products were identified and quantified using the SBSE extraction technique. The submerged cultivation of *Auriporia aurulenta* resulted in the production of several volatile compounds in different concentrations, influenced by the use of umbu, cajá, persimmon, and plum residues as substrates in the fermentation medium. The substrate containing umbu residue stood out from the others and enabled the expressive production of bioaromas with pleasant odors, with predominantly floral and sweet aromatic notes, with an emphasis on the compounds 2-phenethyl acetate and 2-phenylethanol. The highest production of these compounds was obtained after 3.5 days of fermentation in acid medium, and after 7 days in basic medium, respectively. The variations in the residue and pre-inoculum concentrations showed that higher concentrations of 2-phenethyl acetate were obtained with an increase in umbu residue, while an increase in pre-inoculum significantly influenced the increase in the 2-phenylethanol production. In addition to these results, the umbu residue also favored the formation of the compounds 3-(*Z*)-hexenyl acetate, cuminyl acetate, (*E*)-$\beta$-damascenone, *p*-cymen-7-ol, 1,4-*p*-menthadien-7-ol, and (*E*)-2-hexenyl acetate. It is also worth mentioning the production of the compounds benzaldehyde, cinnamyl acetate, cinnamyl alcohol, octyl acetate, and hexyl acetate using cajá residue and 1-octenyl-3-acetate using persimmon residue. The basidiomycete *A. aurulenta* successfully transformed agroindustrial residues into mixtures of complex and highly interesting natural aromas.

**Author Contributions:** Conceptualization, R.D.D.S.; methodology, R.D.D.S. and M.S.D.J.; software, R.D.D.S.; formal analysis, R.D.D.S. and R.A.R.D.S.; investigation, R.D.D.S.; resources, N.N.; data curation, R.D.D.S., R.A.R.D.S., M.S.D.J., H.C.S.A. and J.P.N.; writing—original draft preparation, R.D.D.S. and N.N.; writing—review and editing, M.T.S.L.N. and N.N.; visualization, N.N.; supervision, N.N.; project administration, N.N.; funding acquisition, N.N. All authors have read and agreed to the published version of the manuscript.

**Funding:** This research was funded by CNPq (Conselho Nacional de Desenvolvimento Científico e Tecnológico), Brazil, vide research project Instituto Nacional de Ciência e Tecnologia de Frutos Tropicais (Project 465335/2014-4) in developing this work, CAPES (Coordenação de Aperfeiçoamento de Pessoal de Nível Superior; Financial code 001), Brazil, for their fellowships.

**Institutional Review Board Statement:** Not applicable.

**Informed Consent Statement:** Not applicable.

**Data Availability Statement:** Data available on request.

**Conflicts of Interest:** The authors declare no conflict of interest.

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
