# Peer review of "The Production of Bioaroma by Auriporia aurulenta Using Agroindustrial Waste as a Substrate in Submerged Cultures"

_fermentation, doi:10.3390/fermentation9070593_

Round 1

Reviewer 1 Report

The manuscript presents novel information, of interest to Journal readers, however it cannot be published until the manuscript is reorganized and all sections are clearly displayed. There is an inconsistency between the title and the content of the article, and that the entire manuscript deals with the production of compounds that provide aroma or odor, while the title indicates the production of bio-flavors. There must be clarity and justification about the experimental design, for example, explain why there is a period of 7 days in the submerged fermentation, and the reason for determining that the optimization would be carried out with sampling every 3.5 days, indicate the reason why the factorial design considered the pre-inoculum and the residue (from what is seen, the residue by weight was considered, but which residue is not indicated, since there are 4), it is suggested that a table be shown with all the treatments of the design factorial, and explain how the levels were obtained, since in the table 1 the 0 levels do not coincide with the values used in the previous fermentation. The results must be shown in an organized way, a relationship between the sequence of experiments and interpretation of results is not observed. Apparently, table 2 shows volatile compounds produced in the 7-day fermentation, on the other hand, the results obtained in the optimization kinetics should not present the justification of what was determined, since that should be presented in the methodology. Line 266 indicates that the umbu showed small amounts of sugars, however, the methodology does not indicate that this determination has been made in the agroindustrial residues, or it refers to the culture medium with that residue, it should be clarified. Figure 1, where was it obtained from? Nowhere in the methodology is the determination of umbu sugars indicated, or is it from the culture medium with that residue? And the pH values 3 and 6, are they from the culture medium or from the determination of sugars? These determinations are not indicated in the methodology. Figure 2 is also described at two pHs, it is not understood what was done, clarify the methodology.

Correct grammatical errors, verify the way of citing (for example line 103), the Miller reference is not in the list and must be cited correctly.

None

Author Response

Detailed response to Reviewers comments

Ms. Ref. No. Fermentation-2442303

General Comments:

We appreciate once again the comments made by the Reviewers and thank them all for their valuable suggestions. The Reviewers have brought up some good points and we wish to inform that all these comments have been duly considered in the preparation of new corrected manuscript.

Reply on Reviewers comments:

Reviewer´s comments are reproduced below in Normal format while our reply & rebuttal is presented in blue color text. In order to facilitate the reply on comments raised by the Reviewers, each reply is fully submitted in this text in spite of its citation at various times.

COMMENTS TO THE AUTHOR:

Reviewer #1:

There are some comments for the authors:

1 - There is an inconsistency between the title and the content of the article, as the entire manuscript deals with the production of compounds that provide aroma or odor, while the title indicates the production of bioflavors.

  • Comment was taken into account and the term “bioflavor” was replaced by the term “bioaroma” throughout the Manuscript.

2 - There must be clarity and justification about the experimental design, for example, explain why there is a period of 7 days in the submerged fermentation, and the reason for determining that the optimization would be carried out with sampling every 3.5 days, indicate the reason the factorial planning considered the pre-inoculum and the residue (as you can see, the residue by weight was considered, but which residue is not indicated, since there are 4), it is suggested that a table be presented with all the treatments of the design factorial, and explain how the contents were obtained, since in Table 1 the 0 contents do not coincide with the values used in the previous fermentation. The results must be presented in an organized manner, with no relationship between the sequence of experiments and the interpretation of results being observed. Apparently, table 2 shows the volatile compounds produced in the 7-day fermentation, on the other hand, the results obtained in the optimization kinetics should not present the justification of what was determined, as this should be presented in the methodology.

  • Initially, a 7-day submerged fermentation with aurulenta was carried out in order to know the capacity of this basidiomycete for the production of aroma compounds using each of the agro-industrial residues (umbu, cajá, persimmon and plum) as substrates. The initial results showed that fermentation with umbu residue stood out in relation to other residues in terms of the number and quantity of compounds produced, with emphasis on the compounds viz. 2-phenethyl acetate, 2-phenylethanol and (E)-β-damascenone. Then, a study of the production kinetics of these compounds was carried out for 21 days, with monitoring every 3 and a half days, in order to verify the favorable fermentation time for increasing the production of each of these compounds, using the residue of umbu as substrate, since it favored this production. After determining the viable fermentation time, an optimization study of the pre-inoculum and residue variables was carried out, in order to establish the best conditions to increase the production of the mentioned compounds, aiming at the use of a greater amount of residue and a smaller amount of pre-inoculum. In order to clarify the relationship of kinetics and optimization of production of volatile compounds, the following excerpts were rewritten to the Manuscript:

Pg 3: “The kinetic study of the fermentation process for the production of the main aromatic compounds, using the residue with the highest production of these compounds as substrate, was carried out under the same conditions as in the previous item, being monitored for 21 days with analysis being done on every three and a half days (0; 3.5; 7; 10.5; 14; 17.5 and 21 days), using initial pH 6 and 3. The reduction of total sugars was also monitored in these time intervals, using the method described by Miller [27].

The optimization of fermentation conditions was investigated on the days of highest production, determined in the kinetic study. The effects of the pre-inoculum (X) and residue (Y) quantitative variables were evaluated using a Rotational Central Compound Design (DCCR), consisting of a 22 factorial design, 4 axial points and 3 central points replications, totaling 11 attempts. Table 1 shows the range of variables studied and the corresponding coded levels. In order to maximize the use of residue and minimize the amount of pre-inoculum from the previous fermentation, the values of 6.25 g and 25 mL were considered for the variables X (-1.41) and Y (+1.41), respectively.”

3 - Line 266 indicates that the umbu had small amounts of sugars, however, the methodology does not indicate that this determination was made in agro-industrial residues, or refers to the culture medium with this residue, which must be clarified. Figure 1, where was it obtained from? Nowhere in the methodology is the determination of umbu sugars indicated, or is it from the culture medium with this residue?

  • Figure 1 lists the reduction in total sugars during the 21 days of submerged cultivation of auruenta using umbu residue as a fermentation substrate.
  • The comment was taken into account and the following excerpts were added to the Manuscript:

Pg 2: “Total reducing sugars were determined for the 4 residues using the DNS method [27].”

Pg 8: “The umbu residue showed a small amount of total sugars (0.17 g/L) so that the sugar available in the fermentation process refers to that present in the SNL-minimum, which was reduced in a non-linear way during the fermentation process."

The following reference has been added:

[27] Miller, G.L. Use of dinitrosalicylic reagent for determination of reducing sugar. Anal. Chem. 1959, 31, 426-428. https://doi.org/10.1021/ac60147a030.

4 - And the values of pH 3 and 6, are they from the culture medium or from the determination of sugars? These determinations are not indicated in the methodology. Figure 2 is also described in two pHs, it is not understood what was done, clarify the methodology.

  • The kinetics of the fermentative process of formation of aroma compounds was evaluated at 2 initial pHs, adjusted to 3 and 6. Thus, the values of pH 3 and 6 refer to the culture medium containing the microorganism and the residue. This methodology was added in answer 2, according to the new rewritten excerpt.
  • The comment was taken into account and the rewriting of this excerpt was done to the Manuscript:

Pg 3: “In sterile SNL broth pH 6 determined in a pHmeter (Hanna, model HI2210), a 1x1 cm portion of the microorganism was dispersed using an Ultra Turrax (Heidolph, model SilentCrusher M). The pre-inoculum was kept in a shaker incubator until development, for 15 days, at 24 °C and 150 rpm, protected from light.”

5 - Correct grammatical errors, check the citation form (for example line 103), the Miller reference is not in the list and must be cited correctly.

  • The suggestion was accepted and the whole manuscript was revised by a native English speaker.
  • The comment was taken into account and corrections were made as follows:

Pg 2:  Miller [27]

Pg 3: Bosse et al. [15]; Grosse et al. [28]; Miller [27]

Reviewer 2 Report

This manuscript describes the study of the nature and quantity of volatile compounds produced from the fermentation of natural extracts with Auriporia aurulenta in submerged fermentation cultures. The total amount of sugars present in the fermentation cultures over time was also reported. This manuscript provides reasonable background information and justification for the research, but this reviewer feels that the introduction could be more detailed. The discussion section should also provide a more thorough explanation for the results observed. The materials and methods section is sufficiently descriptive and detailed. Overall this manuscript is well written, and this reviewer recommends publication upon the implementation of the suggestions listed below.

This reviewer believes that the introduction should be more detailed, providing deeper discussion on the practical value of this work (in particular on the utility of the primary compounds produced in this study), as well as some further discussion on why this work is environmentally advantageous.

The authors should provide some comment as to why the umba residue proved to be the most productive residue in this study, with respect to the number of distinct volatile compounds produced and with respect to quantity of volatile compounds produced.

Introduction, Line 33: “industry” should be “industries”

Materials and Methods, Section 2.1, Lines 71-75: A long list of chemicals is provided as an incomplete sentence, without stating where these compounds were obtained.

Table 2: “(Z)3-hexenyl acetate” should be “(Z)-3-hexenyl acetate”

Figures: In some figures, “d” is used to signify “days, and in other cases “days” is used. Also, in some cases “cultivation time” is used, and in other just “time”. Both should be standardized.

Figure 2 caption: “A. aurulenta” should be italicized.

Table 3: I believe “essay” should be “assay”

Table 3: Units need to be provided for the values listed

Results, Section 3.3, Line 327: I believe “(10g)” should be “(20g)”

Results, Section 3.3, Line 353: “(R2)” should be “(R2)”

Author Response

Detailed response to Reviewers comments

Ms. Ref. No. Fermentation-2442303

General Comments:

We appreciate once again the comments made by the Reviewers and thank them all for their valuable suggestions. The Reviewers have brought up some good points and we wish to inform that all these comments have been duly considered in the preparation of new corrected manuscript.

Reply on Reviewers comments:

Reviewer´s comments are reproduced below in Normal format while our reply & rebuttal is presented in blue color text. In order to facilitate the reply on comments raised by the Reviewers, each reply is fully submitted in this text in spite of its citation at various times.

COMMENTS TO THE AUTHOR:

Reviewer #2:

1 - This reviewer believes that the introduction should be more detailed, providing a deeper discussion of the practical value of this work (in particular the usefulness of the primary compounds produced in this study), as well as a more in-depth discussion of why this work is environmentally advantageous.

  • Suggestion accepted and the following excerpt was added in the Manuscript:

Pg 2: “With the growing increase in the generation of agroindustrial residues from the production and processing of fruits and with the concern for the preservation of the environment, alternatives for the sustainable disposal of these materials have been largely investigated and one of them is the use of these residues as culture medium for microorganisms producing volatile aromatic compounds [25,26].”

The following references have been added:

[25] Chattopadhyay, P.; Banerjee, G.; Sen, S. K. Cleaner production of vanillin through biotransformation of ferulic acid esters from agroresidue by Streptomyces sannanensis. J. Clean. Prod. 2018, 182, 272-279. https://doi.org/10.1016/j.jclepro.2018.02.043.

[26] Sharma, A.; Sharma, P.; Singh, J.; Singh, S. et al. Prospecting the potential of agroresidues as substrate for microbial flavor production. Front. Sustain. Food Syst. 2020, 4, 1-11. https://doi.org/10.3389/fsufs.2020.00018.

2 - The authors should comment on why umbu residue proved to be the most productive residue in this study, with respect to the number of different volatile compounds produced and with respect to the amount of volatile compounds produced.

  • We appreciate the comment. However, the present manuscript demonstrated the potential of agro-industrial residues as substrates in submerged fermentation processes for the production of bioaromas. In fact, the umbu residue was the most promising in terms of the number and quantity of compounds produced. Subsequent work will focus on the determination of precursors and metabolic pathways for the formation of 2-phenylethanol, 2-phenethyl acetate, among others.

3 - Introduction, Line 33: “industry” should be “industries”

  • Suggestion accepted and correction made on Pg. 1.

4 - Materials and Methods, Section 2.1, Lines 71-75: A long list of chemicals is given as an incomplete sentence, without indicating where these compounds were obtained.

  • The comment was taken into account and corrections were made:

Pg 2. “Glucose was purchased from Neon Commercial Analytical Reagents Ltd (Suzano, São Paulo, Brazil); Potassium hydrogen phosphate was purchased from Qhemis, Scientific Hexis (Jundiaí, São Paulo, Brazil); magnesium sulfate, zinc sulfate and eth-ylenediaminetetraacetic acid (EDTA) were purchased from Dinâmica Química Contemporânea Ltd (Indaiatuba, São Paulo, Brazil); copper sulfate pentahydrate was acquired from Chemco Industry and Commerce Ltd (Hortolândia, São Paulo, Brazil); 3,5-dinitrosalicylic acid (DNS) and ferric chloride were purchased from Vetec Fine Chemicals Ltd (Duque de Caxias, Rio de Janeiro, Brazil); Mueller Hinton Agar was purchased from Kasvi Import and Distribution of Products for Laboratories Ltd (São José dos Pinhais, Paraná, Brazil);”

5 - Table 2: “(Z)3-hexenyl acetate” must be “(Z)-3-hexenyl acetate”

  • Suggestion accepted and correction made in Table 2.

6 - Figures: In some figures, “d” is used to mean “days”, and in other cases, “days” is used. Also, in some cases “cultivation time” is used and in others just “time”. Both must be standardized.

  • Suggestion accepted and corrections made in Figures 1 and Figure 2.

The term ‘days’ has been used instead of d, and cultivation time instead of time, throughout the Manuscript.

7 - Caption of Figure 2: “A. aurulenta” should be italicized.

  • Suggestion accepted and correction made in the legend of Figure 2.
  • Pg 10: “Figure 2. Kinetics of 2-phenethyl acetate (A), 2-phenylethanol (B) and (E)-β-damascenone (C) production during submerged cultivation of aurulenta with umbu residue at pH 3 and pH 6.”

 8 - Table 3: I believe “essay” should be “assay” and units need to be provided for the listed values

  • Suggestion accepted and correction made in Table 3 of the Manuscript.

Table 3. Effect of pre-inoculum and residue concentrations on the production of volatile compounds obtained in submerged fermentation with A. aurulenta.

Experimental Values

Predicted Values

Assay

Pre-inoculum (mL)

Residue (g)

2-Phenethyl acetate (mg/L)

2-Phenethyl alcohol (mg/L)

(E)-ß-Damascenone (mg/L)

2-Phenethyl acetate (mg/L)

2-Phenethyl alcohol (mg/L)

1

11.44

9.04

7.32

5.41

0.19

4.00

4.80

2

21.56

9.04

3.57

6.71

0.18

4.00

8.93

3

11.44

17.22

13.05

2.86

0.23

13.43

4.80

4

21.56

17.22

9.28

6.25

0.47

13.43

8.93

5

8.00

13.12

8.25

8.22

0.24

5.78

3.97

6

25.00

13.12

10.10

15.56

0.23

5.78

10.91

7

16.5

6.24

4.23

6.65

0.23

6.14

6.55

8

16.5

20.00

24.47

9.90

0.19

21.96

6.55

9

16.5

13.12

4.10

4.15

0.22

5.78

6.55

10

16.5

13.12

4.11

4.67

0.24

5.78

6.55

11

16.5

13.12

3.41

4.68

0.26

5.78

6.55

9 - Results, Section 3.3, Line 327: I believe “(10g)” should be “(20g)”

  • Suggestion accepted and correction made on Pg 12 of the Manuscript.

10 - Results, Section 3.3, Line 353: “(R2)” should be “(R2)”

  • Suggestion accepted and correction made on Pg 14 of the Manuscript.
